# Analysis of Uncertainties in Inductance of Multi-Layered Printed-Circuit Spiral Coils

**DOI:** 10.3390/s22103815

**Published:** 2022-05-18

**Authors:** Myounggyu Noh, Thien Vuong Bui, Khanh Tan Le, Young-Woo Park

**Affiliations:** Department of Mechatronics Engineering, Chungnam National University, Daejeon 34134, Korea; sukhobui@gmail.com (T.V.B.); lekhanhtan29@gmail.com (K.T.L.); ywpark@cnu.ac.kr (Y.-W.P.)

**Keywords:** PCB coil, inductance, uncertainty, eddy-current sensor

## Abstract

Eddy-current sensors are widely used for precise displacement sensing and non-destructive testing. Application of printed-circuit board (PCB) technology for manufacturing sensor coils may reduce the cost of the sensor and enhance the performance by ensuring consistency. However, these prospects depend on the uniformness of the sensor coil. Inductance measurements of sample coils reveal rather considerable variations. In this paper, we investigate the sources of these variations. Through image analysis of cut-away cross-sections of sensor coils, four factors that contribute to the inductance variations are identified: the distance between layers, the distance between tracings, cross-sectional areas, and misalignment among layers. By using and extending existing method of calculating inductance of spiral coils, the inductance distributions are obtained when these factors are randomly varied. A sensitivity analysis shows that the inductance uncertainty is most affected by the uniformness of the spacings between coil traces and the distances between layers. Improvements in PCB manufacturing process can help to reduce the uncertainty in inductance.

## 1. Introduction

Eddy current sensors are widely used for precision measurement of displacement without mechanical contact. Similar to proximity and inductive sensors, eddy-current sensors are capable of working in hostile environments where dust or liquids may appear in the gap between the sensor and the target [1,2]. A coil is placed at the core of the sensor, which plays an essential role in generating an electromagnetic field as an excitation field for the sensor. The working principle of the sensor is the interaction between the excitation field and the induced field in the target due to eddy currents. Typically, the coil is wound around a ferromagnetic core for flux focusing [3]. Due to space constraints, it is very difficult to maintain the consistency of winding in a tiny sensor. This is the main reason that small form-factor eddy-current sensors are expensive.

Printed-circuit board (PCB) is a mature technology that is extensively used in the industry. Implementation of spiral coils using PCB is attracting interests because it is possible to mass-produce the coil cost-effectively [4]. In addition, inductance-to-digital converters (LDC) have been commercialized [5], which simplifies signal processing and transmission. Inductance approximation for a flat PCB coil can be obtained using a modification of Wheeler’s expression with high accuracy (2–3% of error) [6,7]. However, the feasibility of using PCB coils for eddy-current sensors also depends on the consistency of the spiral coils. If there are considerable variations in the inductances of PCB coils, the sensor accuracy would be limited.

In this work, we measure the inductances of spiral coils implemented with PCB technology and have observed fairly large variations in inductance. Defining what factors contribute to the inductance variations and how the manufacturing process of PCB affects the uncertainties are therefore necessary. Through the analyses of images taken from the cutaway sections of PCB containing spiral coils, four geometric factors contributing to inductance variation are identified and their variabilities are quantified. To see how much these geometric variations result in inductance uncertainty, the methods of calculating the inductance of multi-layer PCB coils are investigated. Existing method such as [8] can handle most of the factors. However, none of the existing methods can deal with the case of misaligned layers. By utilizing the method of calculating the mutual inductance between two misaligned filament [9], we extend it for the case of misaligned coils. Similar to Monte-Carlo simulations, the distributions of inductance due to random variations in geometric factors are obtained. A sensitivity analysis is carried out to see what factors contribute most to the inductance uncertainty.

This paper is organized as follows. Following the Introduction, inductance measurement of the coil samples is presented. To assess the inductance variations among coils, the measurement uncertainty is identified to separate the effect of geometric factors. Next, the results of measurements on cut-away images are described. Four geometric factors are identified that contribute most to the inductance uncertainty. The amount of variations in these factors are described in this section. Section 5 on uncertainty analysis discusses the method of calculating the inductance of spiral coils and the inductance distributions while the geometric factors are randomly varied. A sensitivity analysis is carried out to quantify how much the geometric factors contribute to the inductance uncertainty. It also discuss the issues of PCB manufacturing process that are the causes of geometric variations.

## 2. Variations in Inductance of PCB Coils

### 2.1. Coil Samples and Inductance Measurements

To study the feasibility of using PCB coils for eddy-current sensors, sample coils are manufactured. The coil is defined by the inner diameter ($d_{i}$), outer diameter ($d_{o}$), the number of turns per layer (*n*), the number of layers (*l*), trace width (*w*), and trace spacing (*s*). The schematic and picture of the sample coil are shown in Figure 1 and the specifications of the coil are listed in Table 1. The inductance measurement is carried out with a total of 15 coils.

The coil can be modeled as a series combination of inductance ($L_{s}$) and resistance ($R_{s}$). If a capacitor ($C_{p}$) is added in parallel with the sensor coil, the sensor impedance can be expressed as
(1)$$Z_{s}\left(\omega \right)=\frac{R_{s}+j\omega L_{s}}{1-\omega^{2}L_{s}C_{p}+j\omega R_{s}C_{p}}\;,$$
where *j* is the imaginary number and ω is the excitation frequency. If the *Q* is high, the effect of resistance can be ignored. Then, the self-resonant frequency (SRF) is approximated as
(2)$$f_{\mathrm{SRF}}=\frac{1}{2\pi \sqrt{L_{s}C_{p}}}\;.$$

Since SRF is dependent on the inductance, the measurement of SRF is equivalent to the inductance measurement. The presence of the field induced by eddy currents changes inductance and SRF in turn [10], which is the principle of eddy-current sensors.

The inductance of the sample coils is indirectly measured with an impedance analyzer (E4990A, Keysight Technologies, Colorado Springs, CO, USA). The range of the excitation frequency is from 100 kHz to 4 MHz. The SRF is typically around 1 MHz. A curve-fitting procedure using least-square-error (LSE) minimization is employed to identify the self-resonant frequency and then the inductance is estimated by (Equation 2). By repeating the measurement 20 times of each coil, a set of 300 values of inductance are obtained.

### 2.2. Inductance Variation

If the inductance measurements are variable, two sources can contribute to the variability: one being the coil itself and the other the uncertainty of the measurement setup. If the coils are not consistent due to manufacturing processes, the inductances would change from sample to sample (inter-sample variation). To quantify this inter-sample variation, the uncertainty of the measurement setup must be identified first.

The inaccuracy of the measurement process can be estimated by repeating the measurements for the same coils. For all 15 sample coils, the measurements are repeated 20 times. The standard deviations of the measurements are calculated. Then, the measurement uncertainty is obtained from
(3)$$\mathrm{Measurement}\;\mathrm{Uncertainty}=\frac{\sigma_{L}}{{\left(\Delta L\right)}_{\mathrm{FS}}}\;,$$
where $\sigma_{L}$ is the standard deviation of the coil under test, and ${\left(\Delta L\right)}_{\mathrm{FS}}$ is the inductance change in the full sensing range.

Table 2 summarizes the variations in the measured inductances. The average and the standard deviation of all measurements are 54.2 μH and 142 nH, respectively. For each coil, the standard deviation is calculated, The maximum from 20 samples is reported as the maximum intra-sample variation, which is 2 nH. The difference between the largest and the smallest average inductances of 15 coils is shown as the maximum inter-sample variation (550 nH).

If the coil is used as a position sensor, the inductance change in the full sensing range is at most 10% of the base value. Therefore, the measurement uncertainty is quite small: 0.04% for the worst case. We can practically disregard the intra-sample variation.

The changes in inductances from one coil to another are much greater than the measurement uncertainty. The average inductances of 15 coils range from 53.92 μH to 54.47 μH. Based on the maximum inter-sample variation, the variability is 10.1%. The inductance change can of course be calibrated. However, additional calibration increases the overall complexity and cost of the sensor and mitigates the advantage of using PCB techniques.

### 2.3. Possible Sources of Inductance Variations

Why is there so much variability? What are the causes of inconsistencies in parameters of the sensing coils? Which stage in the PCB manufacturing process do these inconsistencies originate from? To answer these questions, the sensor board is cut in half, and the cut-away sections are examined. The cross-sections of coil tracings in these cut-away sections are fairly irregular.

We have identified four different types of irregularities, which are shown in Figure 2, Figure 3, Figure 4 and Figure 5. An ideal layout of four-layer PCB coils is illustrated as Figure 6 where the distances between layers are uniform and the tracings of one layer are perfectly aligned with those of other layers. Each tracing also has uniform areas (width and thickness). In reality, the distances between layers are not uniform, as shown in Figure 2. This variability is unavoidable due to the nature of the PCB manufacturing process. Thus, assessing the contribution of this inter-layer distance to the variability of coil inductance is also necessary. The distances between tracings are not uniform, either, as displayed in Figure 3. Also, the cross-sectional areas of tracings are quite variable, in which the thicknesses of tracings are not changing much, but the widths show a significant variation as illustrated in Figure 4. Besides, the center lines of layers do not coincide as well, causing a layer-to-layer misalignment as described in Figure 5. Figure 7 shows the actual cut-away section, where all four irregularities are observable.

In this paper, an image processing technique is employed to quantify the variabilities of these three factors from the cut-away images of all 15 coils.

## 3. Image Analysis

### 3.1. Procedure for Image Capture

As mentioned above, the sensor board is cut in half. The halved board is positioned so that the cut-away sections are directly under a microscope camera (iMegascope 1080P, Sometech, Seoul, Korea) with a magnification of 160. In order to maximize image resolution, the cut-away section is divided into eight subsections. Therefore, a total of 120 images are taken for 15 coils. The photographing process is set up as Figure 8 where the green background is used make the images of the tracings stand out.

### 3.2. Image Processing

The goal of image processing is to identify the copper tracings and obtain information regarding the tracings such as the center coordinates, the areas (in terms of width and thickness), and misalignments. The raw images are first converted into binary images (black and white images) by computing the global threshold using Otsu’s method. After the segmentation, the white color represents copper tracings and the background should be black, which identifies the objects to be measured [11]. However, as shown in Figure 9, some sporadic white grains appear where copper tracings do not exist and some tiny black points are present inside the copper tracings, which can affect the results of irregularities measurement. By implementing a flood-fill operation on the background pixels and applying a 3-by-3 neighborhood median filtering repeatedly, these salt and pepper noises can be eliminated. On the other hand, the tracings cut by the edges of images must be removed, as they would be falsely identified as irregularities. Since the white regions are then converted into objects and information such as center coordinates as well as bounding box properties can be extracted, the objects cut by the edges of the image are determined. All of the needed functions mentioned above are provided by MATLAB Image Processing Toolbox. The end of the process results in noise-free images, which are shown as a sample in the right image of Figure 9.

## 4. Measurement Results

Through the image analysis described in the previous section, the variabilities of coil parameters are measured in terms of four uncertain sources. The first is the trace spacing. The design value of the spacing is 102 μm, but the average of the tracing spacings are measured to be 140 μm (37% larger than the design). Especially, the tracings of the outer-most layers are deformed much and have irregular spacings. The standard deviation of the spacing is 32.3 μm, which is 23.1% of the average value.

While the tracing thicknesses are relatively uniform, the trace widths are not. The average value is 83 μm, which is much smaller than the design of 102 μm. The standard deviation is 8.6 μm, which is 10.3%. The inter-layer distances are quite different from the ideal case of Figure 6. The distance between layers 2 and 3 is much larger than those between 1–2 and 3–4 layers. The distance between the two inner layers, $d_{23}$, has an average of 966 μm, which is quite uniform because the standard deviation is less than 1% of the average. The averages of $d_{12}$ and $d_{34}$ are 139 μm and 137 μm, respectively. These inter-layer distances are also quite uniform: standard deviations are 3.5% and 3.9% of the respective averages.

The misalignment between layers have significant variability. Ideally, the misalignment must be zero, but the average is 93 μm. Furthermore, the standard deviation of misalignment is 27.5 μm, which is close to 30% of the average. Table 3 summarizes all the measurements obtained.

## 5. Uncertainty Analysis

### 5.1. Calculation of PCB Coil Inductance

In order to assess how much the coil inductance varies due to the irregularities described in the previous section, we need to be able to calculate the inductance of the PCB coil. However, an accurate calculation of the PCB coil inductance is not a straightforward task when the coil patterns are irregular. For a coaxial pair of circular filaments having the radii of *a* and *r*, and axially displaced by *z*, Maxwell derived an expression of mutual inductance as [12]
(4)$$M=\mu_{0}\sqrt{ar}\cdot \frac{2}{f}\left[\left(1-\frac{f^{2}}{2}\right)K\left(f\right)-E\left(f\right)\right]$$
where K(f) and E(f) are complete elliptic integrals of the first and second kind, respectively, and where
(5)$$f=\sqrt{\frac{4ar}{z^{2}+{\left(a+r\right)}^{2}}}\;.$$

Extending (Equation 4) to the case of a coil pair with rectangular cross-sections is rather complex, as it involves an evaluation of an indefinite integral of Bessel functions [8,13]. Numerous approximate methods have been proposed (for example, [14,15,16,17]) In this paper, we used the approximation that replaces the coil with a pair of filaments displaced by “geometric mean distance” (GMD) and using (Equation 4) to compute the inductances between all filaments (Lyle’s method) [18]. The essence of this method is illustrated in Figure 10. For the left coil with $w_{1}<h_{1}$, the coil is replaced with two filaments (1 and 2) with radii r+α and r−α [18,19]:(6)$$r=R\left(1+\frac{w_{1}^{2}}{24R^{2}}\right)$$
(7)$$\alpha =\sqrt{\frac{h_{1}^{2}-w_{1}^{2}}{12}}$$
where *R* is the geometric mean of the inner and outer radii of the coil. If $w_{2}>h_{2}$, as shown in the right of Figure 10, the coil is again replaced with two filaments (3 and 4) having the same radius of *r* at a distance β on either side of the mid-plane of the coil.
(8)$$r=A\left(1+\frac{h_{2}^{2}}{24A^{2}}\right)$$
(9)$$\beta =\sqrt{\frac{w_{2}^{2}-h_{2}^{2}}{12}}$$
where *A* is the geometric mean radius. The mutual inductance between two coils is given by
(10)$$M=\frac{M_{13}+M_{14}+M_{23}+M_{24}}{4}$$
where $M_{ij}$ is computed from (Equation 4) and (Equation 5) with *z* replaced with the geometric mean distance (GMD).

If PCB coils are ideal as illustrated in Figure 6, we can utilize the formula proposed by Mohan et al. [7] to calculate the self-inductance of a single-layer spiral coil, which is given as
(11)$$L=\frac{\mu_{0}N^{2}d_{avg}}{2}\left(\ln\frac{2.46}{\rho }+0.2\rho \right)$$
where $\mu_{0}$ is the permeability of free space, $d_{avg}$ the average diameter, *N* the number of turns, and ρ the fill ratio
$$\rho =\frac{d_{o}-d_{i}}{d_{o}+d_{i}}$$

The mutual inductances between layers can be obtained from the coupling factor obtained from experiments [20]. The total inductance of a multi-layer PCB coils can be calculated by
(12)$$L_{\mathrm{total}}=\sum_{i=1}^{N}L_{i}+2\sum_{j=1}^{N-1}\sum_{m=j+1}^{N}M_{j,m}$$
where $L_{i}$ is the self-inductance of *i*-th layer and $M_{j,m}$ is the mutual inductance between the layers *j* and *m*.

If the pattern of the PCB coils is irregular, it is not possible to use (Equation 12). For the irregularities in Figure 2, Figure 3 and Figure 4, Lyle’s method can be used to calculate the inductance by the procedure illustrated in Figure 11.

The above procedure cannot be applied to the case of layer misalignment (Figure 5), as it assumes that the layers are coaxial. For the pair of circular filaments arbitrarily positioned in space, the method proposed by Babic et al. [9] can be used. This method computes the mutual inductance between two filaments as illustrated in Figure 12.

However, this method is unable to handle the coils with rectangular cross-sections. In this paper, we extended the method of [9] to the case of PCB coils by combining it with Lyle’s method. Therefore, the procedure to calculate the inductance of PCB coils with layer-to-layer misalignment can be summarized in Figure 13.

As summarized in Table 4, the validity of the inductance calculations used in this paper is confirmed by investigating two cases of spiral coils. The first is the four-layer coils presented in [8]. The second is the sensor coil design described in Table 1. For the example case of [8], the results by Lyle’s method are exactly the same as the results by the extended Babic’s method. Reference [8] provides the FEA result as 628 nH, which is very close to our results. However, the result by Mohan’s formula is 44% of the FEA. This means that Mohan’s formula is not suited to a case where the number of turns is small (three, as in the example of [8]).

For the sensor coil design in Table 1, Lyle’s method again produces exactly the same inductance as the extended Babic’s method. Mohan’s formula calculates the inductance close to other methods (about 4% difference). Overall, it is evident that both Lyle’s method and the extended Babic’s method can calculate the inductance of PCB coils accurately. Furthermore, these methods can handle the variations of coil geometries, while Mohan’s formula assumes uniform configuration. It is also noted that the skin effect on coil inductance is negligible, as the calculated inductance is not very difference from the average of measurements (56.0 μH vs. 54.2 μH). Since the skin depth is around 66 μm at the self-resonant frequency of 1 MHz, the cross-sectional area of a single trace is small enough to be influenced by the skin effect.

### 5.2. Assessment of Inductance Variation Due to PCB Irregularities

Using the aforementioned inductance model, it is possible to assess how much the inductance changes due to the irregularities observed in actual PCB coils. First, a reference design is determined, based on the measurements. The width and the spacing of trace are 83 μm and 140 μm, respectively. The inter-layer distances are: $d_{12}=139\;\mu$m, $d_{23}=966\;\mu$m, and $d_{34}=137\;\mu$m. Table 5 compares the inductance by Mohan’s formula, Lyle’ method, extended Babic’s model, and the average of measurements. Lyle’s method and the extended Babic’s model both agree quite well with Mohan’s formula and the average of measurements.

The procedure to assess the inductance variation is as follows. The parameter of interest is randomly varied within one standard deviation from the average value in Table 3. For example, the widths of all 104 traces (26 turns times 4 layers) are randomly varied. Then the inductance is calculated. This is repeated 10,000 times. The distribution of inductances of the 10,000 trials is analyzed. Extended Babic’s model is used for the misalignment case, while Lyle’s method is used for all other cases. It is assumed that the distributions are normal, and also assumed that individual random variation is representative of actual coils. As shown later, one of the distributions is rather skewed, but the effect is not significant. Due to the second assumption, the variability may be underestimated. However, the purpose of this research is to identify the dominant factors contributing to the variability of inductance. Thus, it is important to maintain the same variability for all factors.

Figure 14 is the distribution of inductances when the layer distances are varied randomly. The mean is 56.23 μH, which is slightly larger than the reference inductance calculated by both Lyle’s method and the extended Babic’s method. The standard deviation is 46 nH. Figure 15 shows the distribution when the spacings between traces are varied randomly. The mean is 56.24 μH, while the standard deviation is 59 nH. The inductance variation due to uncertainty in trace width is infinitesimal, as shown in Figure 16. The standard deviation is only 2 nH. The distribution due to misalignment in Figure 17 is somewhat unsymmetrical, since the inductance only increases irrespective of the direction of misalignments. The standard deviation is 18 nH.

From these results, several observations can be made. First, any irregularities increase the inductance. For the four types of irregularities, the average inductance increases to as much as 56.24 μH from the base value of 56.0 μH. It is also apparent that each irregularity type affects the uncertainty in different proportions. Table 6 compares the sensitivity of irregularity types on the inductance variation. The normalized deviation is obtained by the ratio of the standard deviation to the average value using the data in Table 3. The normalized uncertainty is defined as the ratio of standard deviation to the average inductance as shown in Figure 14, Figure 15, Figure 16 and Figure 17. The last column is the ratio of the normalized uncertainty to the normalized deviation. Clearly, the variation of trace width affects the uncertainty very little in inductance. On an absolute scale, the trace spacing affects the inductance variation the most. The most sensitive irregularity type is the layer distance. Image analysis shows that the layer distance is fairly consistent (3.6% variation at most). However, the sensitivity is the largest.

The results of uncertainty simulations can be compared with the measured variation. As shown in Table 6, the layer-layer misalignment is not a dominant factor. Assuming that the nonlinearity in the distribution of misalignment does not affect too much, the total uncertainty is calculated from
(13)$$\sigma_{\mathrm{total}}=\sqrt{\sum_{k=1}\sigma_{k}^{2}}\;.$$

Using the standard deviations of four types of irregularities, the total uncertainty is 0.074 μH, which is less than the standard deviation of the measurements, which is 0.142 μH. This is rather expected. Cutaway images show that irregularity is systematic while random variation is employed in simulations. Another factor contributing to large variations in inductance measurements is the uneven thickness of sensor coil. Due to the manufacturing process of PCB, it is very difficult to maintain the same thickness. As much as 8 μm of difference in thickness across one coil is observed. Since the inductance is sensitive to the distance between the layers, uneven thickness would increase the variability of inductance.

What are the causes of the PCB irregularities? Figure 18 lists two common issues of the PCB manufacturing process [21]. On the left is the undercutting of PCB traces during the etching process. Undercutting is the main cause of the trapezoidal trace cross sections observed in the cutaway images. The etch factor defined as
(14)$$F=\frac{2t}{B-T}$$
can be improved by using heavier copper foil in spite of increased cost. Figure 18b describes the foil outer stack-up for manufacturing multi-layer PCB. A fully-cured C stage is sandwiched between half-cured B stage epoxy. The outermost layers made of copper foil are located outside of this B stage. Then, pressure is applied while heating the platens. This process explains why the thickness of the middle layer ($d_{23}$ is more consistent than $d_{12}$ and $d_{34}$ in Figure 2. While pressurized, the B stages are squeezed much more than the fully-cured C stage. Consistency of layer thickness can be improved if the clad outer stack-up method is used, where only C stages are used.

## 6. Conclusions

The inductance of the sensing coil is directly related to the sensing mechanism of eddy current position sensors. Manufacturing sensing coils using printed-circuit board (PCB) technology can be a solution for implementing high-resolution position sensors at reasonable cost. However, it is observed that the coils from the same batch of PCB exhibit variability in inductance. In this paper, we identified the geometric factors of PCB spiral coils, contributing to the uncertainty of coil inductance: trace spacing, cross-sectional areas of trace, distance between layers, and misalignment of layers. Utilizing and extending existing inductance models, inductance distributions while varying these factors randomly are obtained. The results reveal that the trace spacing affects the most while the layer distance is the most sensitive factor. If the standard process is used for manufacturing PCB coils, the variability in the inductance of the coils is inevitable. It is of course possible to carry out a calibration procedure to account for this variability. For example, the coil inductances are measured with several well-defined and stable gaps between the coil and the target. The measurements will then be used to calibrate the inductance-gap relationship with the added cost of tuning. If the PCB manufacturing process is improved to reduce the inductance uncertainty, this calibration procedure can be eliminated.

## Figures and Tables

**Figure 1 sensors-22-03815-f001:**
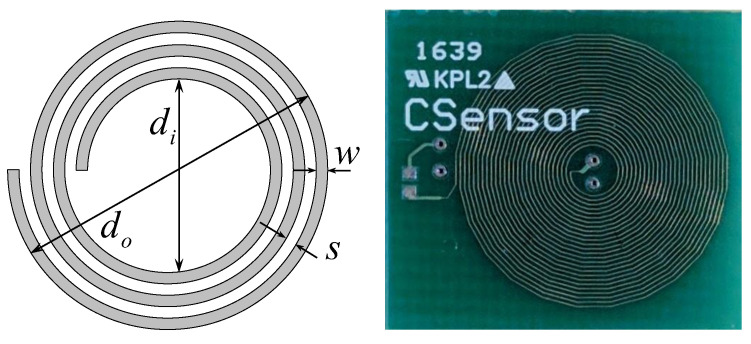
Schematic of sensor coil showing design variables: inner and outer diameter, width, and separation of copper tracing. On the right is the actual sensor coil manufactured using the PCB technique.

**Figure 2 sensors-22-03815-f002:**
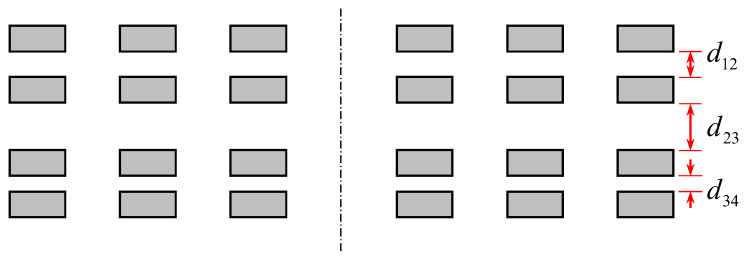
Non-uniform distances between layers.

**Figure 3 sensors-22-03815-f003:**
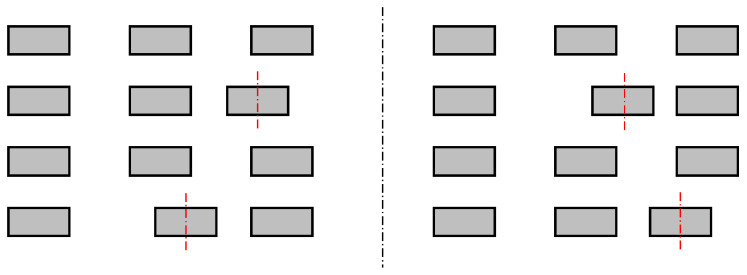
Irregularities in the distances between tracings. Center lines of some traces (red lines) are displaced from the design.

**Figure 4 sensors-22-03815-f004:**
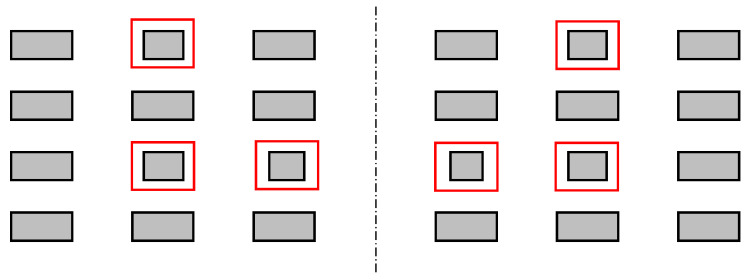
Irregularities in the tracing widths. Some of the traces marked by the red boxes have smaller widths than the design.

**Figure 5 sensors-22-03815-f005:**
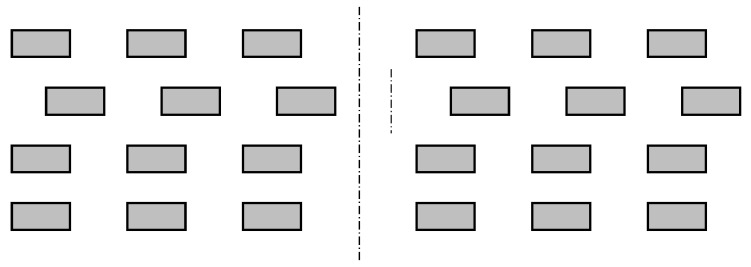
Misalignments of layers.

**Figure 6 sensors-22-03815-f006:**
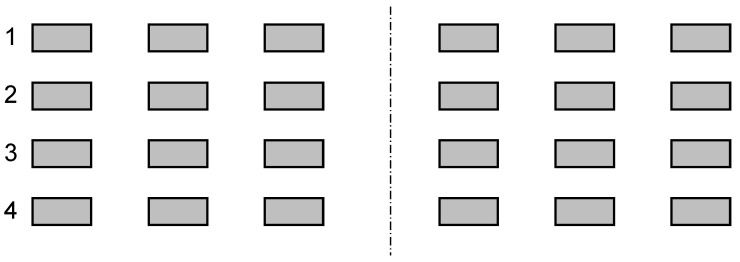
Cross-sectional view of the ideal layout of four-layer PCB sensor coil, with the numbers from 1 to 4 designating the layer.

**Figure 7 sensors-22-03815-f007:**
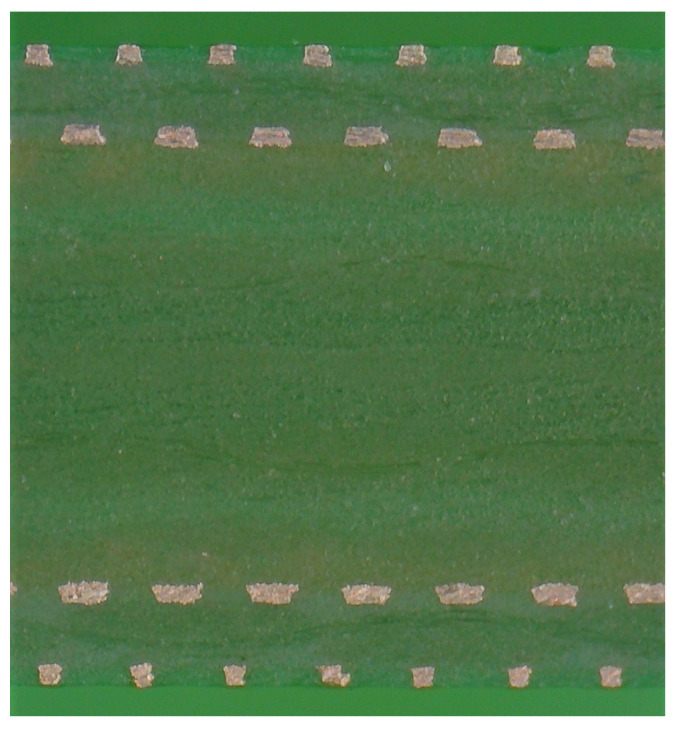
Sample image of a cut-away section of PCB coils.

**Figure 8 sensors-22-03815-f008:**
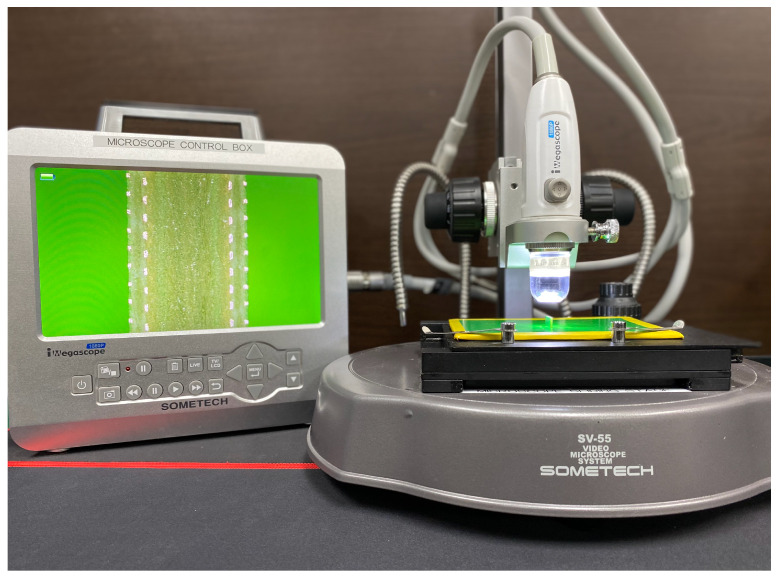
Test setup for image capture.

**Figure 9 sensors-22-03815-f009:**
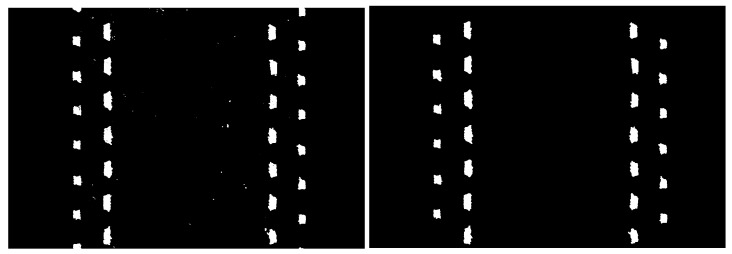
On the left is the image before noise elimination. White specks are visible. On the right is the image after noise elimination.

**Figure 10 sensors-22-03815-f010:**
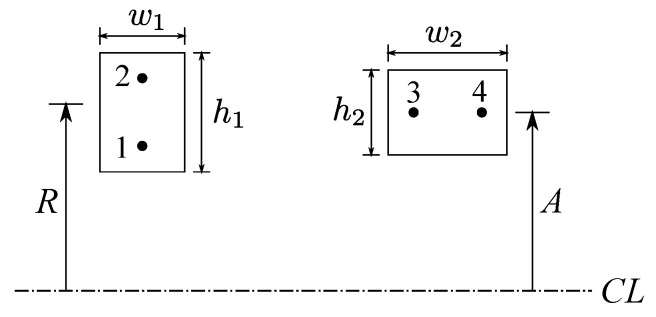
Illustration of Lyle’ method to calculate the inductance of circular coils with rectangular cross-section.

**Figure 11 sensors-22-03815-f011:**
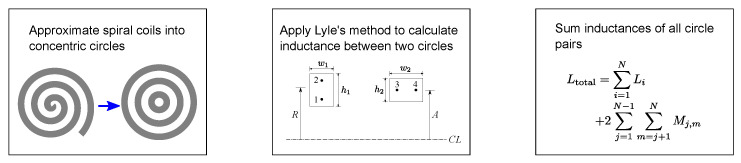
Procedure of inductance calculation using Lyle’s method.

**Figure 12 sensors-22-03815-f012:**
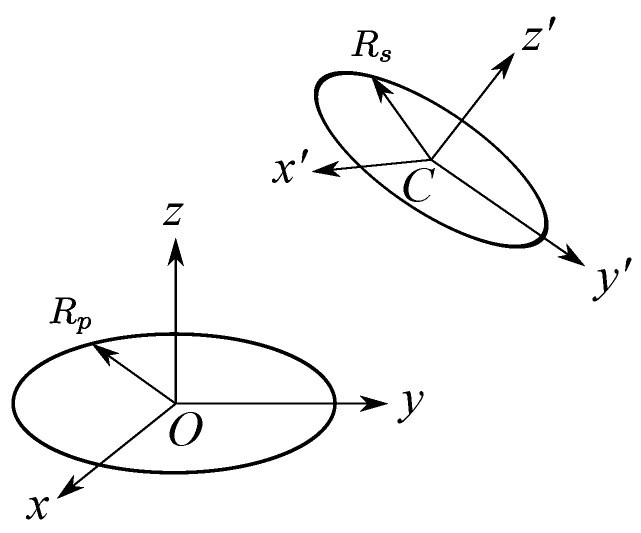
One filament is arbitrarily positioned from the other filament.

**Figure 13 sensors-22-03815-f013:**
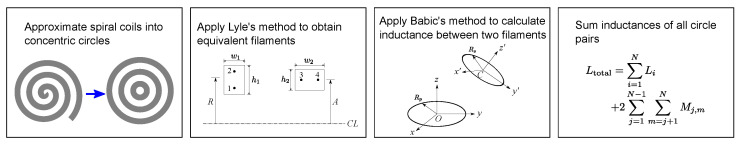
Procedure of inductance calculation using the extended Babic’s method.

**Figure 14 sensors-22-03815-f014:**
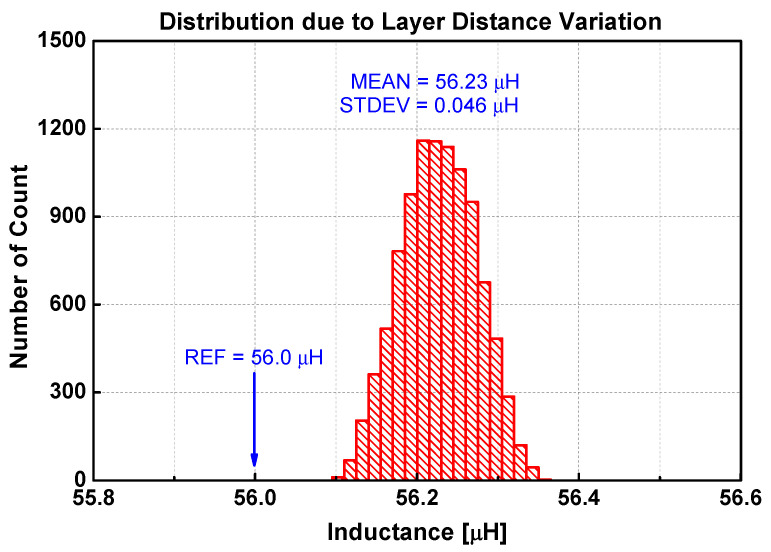
Distribution of inductance when the layer distance is randomly varied 10,000 times.

**Figure 15 sensors-22-03815-f015:**
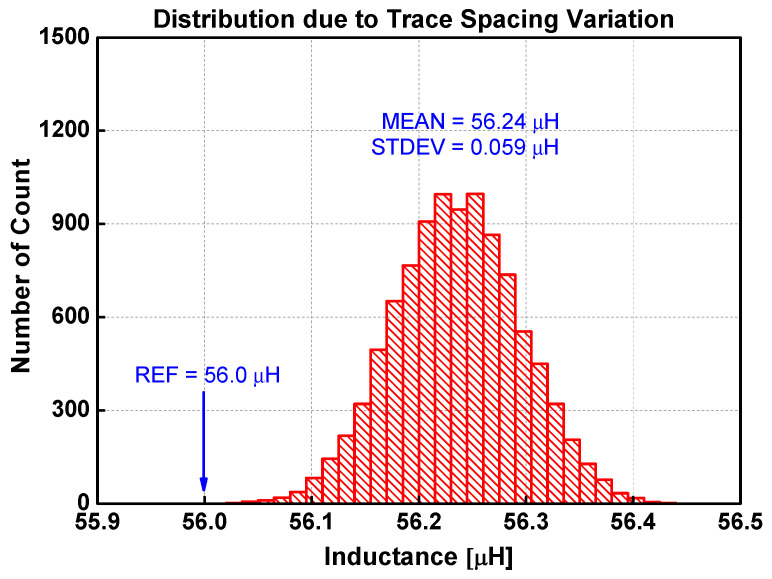
Distribution of inductance when the trace spacing is randomly varied 10,000 times.

**Figure 16 sensors-22-03815-f016:**
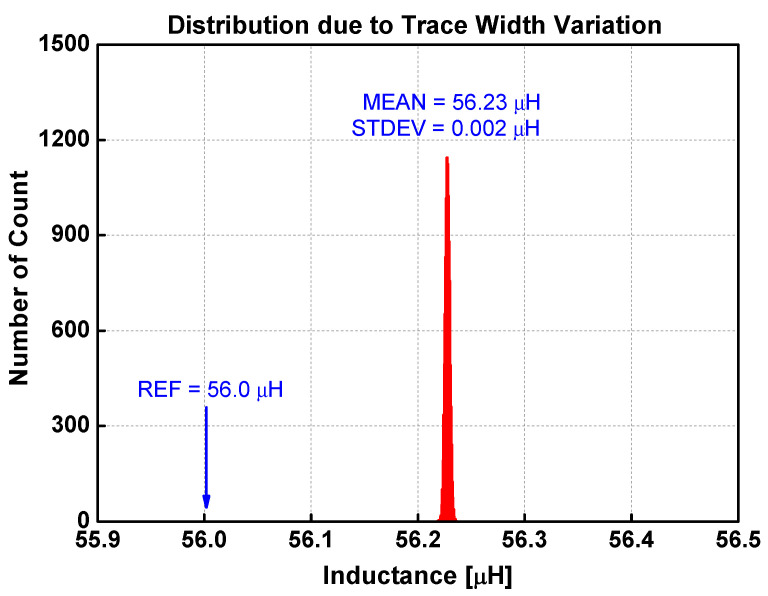
Distribution of inductance when the trace width is randomly varied 10,000 times.

**Figure 17 sensors-22-03815-f017:**
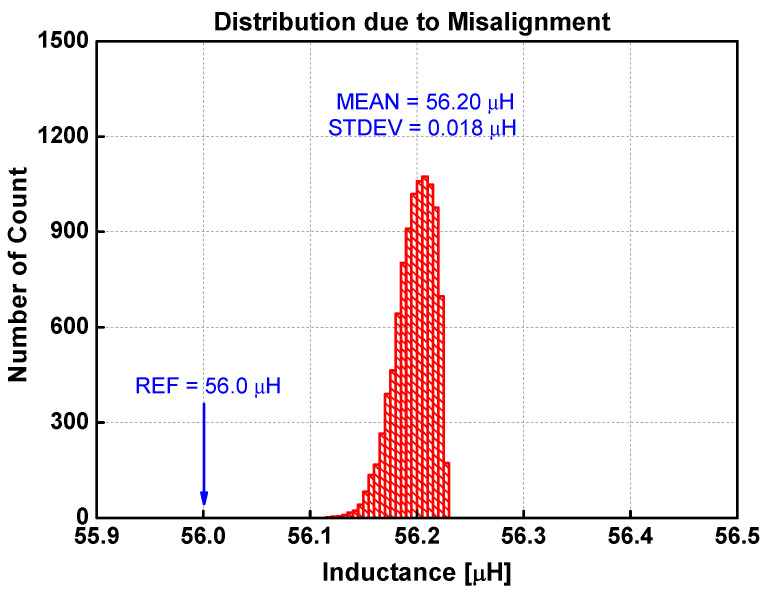
Distribution of inductance when the alignment between layers is randomly varied 10,000 times.

**Figure 18 sensors-22-03815-f018:**
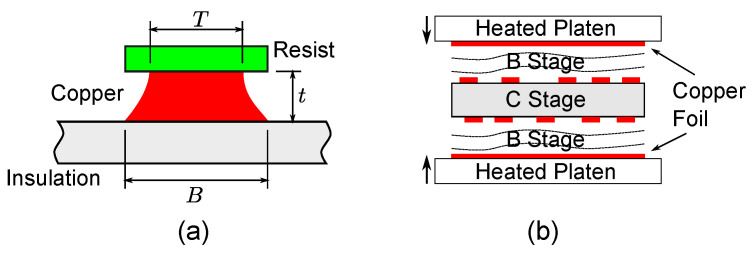
PCB manufacturing process contributing to the inductance variation. (**a**) trace undercutting (**b**) foil outer stack-up.

**Table 1 sensors-22-03815-t001:** Design values for PCB sensor coil.

Parameter	Value
Inner diameter, $d_{i}$ [mm]	2.1
Outer diameter, $d_{o}$ [mm]	12.7
Number of turns per layer, *n*	26
Number of layers, *N*	4
Trace spacing, *s* [mm]	0.102
Trace width, *w*	0.102

**Table 2 sensors-22-03815-t002:** Variations in the inductances of the PCB coil samples (all units are in μH).

Average	54.2
Standard deviation	0.142
Maximum intra-sample variation	0.002
Maximum inter-sample variation	0.550

**Table 3 sensors-22-03815-t003:** Amount of irregularities identified from image analysis.

Irregularity	Mean ± Standard Deviation (μm)
Trace spacing	140 ± 32.3 (23.1%)
Trace width	83 ± 8.6 (10.3%)
Layer distance $d_{12}$	139 ± 4.8 (3.5%)
Layer distance $d_{23}$	966 ± 7.4 (0.8%)
Layer distance $d_{34}$	137 ± 5.4 (3.9%)
Layer-layer misalignment	93 ± 27.5 (29.6%)

**Table 4 sensors-22-03815-t004:** Validations of inductance calculations.

Method	Example in [8]	Sensor Coil
Mohan’s formula [7]	277 nH	49.8 μH
Lyle’s method [8,18]	637 nH	51.8 μH
Extended Babic’s method	637 nH	51.8 μH

**Table 5 sensors-22-03815-t005:** Inductance of reference design calculated by various methods and compared with the average of measurements.

Method	Inductance (μH)
Mohan’s formula	52.9
Lyle’s method	56.0
Extended Babic’s method	56.0
Average of measurements	54.2

**Table 6 sensors-22-03815-t006:** Sensitivity of irregularity parameters on inductance variation.

Irregularity	Normalized	Normalized	
Type	Deviation	Uncertainty	Ratio
Trace spacing	0.231	$9.96\times 10^{-4}$	$0.43\times 10^{-2}$
Trace width	0.103	$0.36\times 10^{-4}$	$0.03\times 10^{-2}$
Layer distance	0.039	$8.18\times 10^{-4}$	$2.10\times 10^{-2}$
Layer-layer misalignment	0.296	$3.20\times 10^{-4}$	$0.11\times 10^{-2}$

## Data Availability

Not applicable.

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
