# Peer review of "Analysis of Uncertainties in Inductance of Multi-Layered Printed-Circuit Spiral Coils"

_sensors, 2022, doi:10.3390/s22103815_

Round 1

Reviewer 1 Report

The paper aims to quantify the impact of uncertainty sources in the inductance of a PCB-pancake-type coil. The investigations are performed at the example of a particular coil design. 

For the first part, a set of coils is manufactured, the inductance quantified with the use of an impedance analyzer and analyzed by destructive testing, to obtain the values for the deviation of inductance and the deviation of the influencing factors.
To quantify the contribution of each influencing factor a model-based approach is used in the second part. Tree approaches for the calculations have been evaluated and to our selected for use in the investigation. The use of two calculation approaches for different parameters (line 249) is considered fine by me, because of the similarity of the results shown in the comparison. The use of a single method for analyzing the influence of all factors would have been preferable.
The calculation approaches were used to obtain the variations in the inductance due to implemented variations in the investigated influencing factors. The procedure of implementing the variations needs to be explained in more detail (see 'must be addressed' for details). The results of this investigation are illustrated in a clean, concise, and understandable way, which is optimized for comparison. (Although I think, I remember there is a standard to use rectangular brackets for the units in axis labels.)
The theoretical calculations reveal a much lower total uncertainty, compared to the measured total uncertainty. It was concluded correctly, that some uncertainty sources could not be represented by the model. 

However, the conclusion section is quite short compared to the length of the paper and the amount of data. If possible it should be, enhanced and improved for a higher benefit of the reader (also see 'specific comments').

In summary, the topic of the paper seems to be very particular but addresses a common problem in the application of eddy current sensors. The results of the numerous investigations and the concise summary of the results provide a valuable contribution to this field.

Must be addressed:

  • Homogeneous variation of parameter or individual random? It is not clear if each layer distance was varied randomly individually or if they have a common random value. (line 245)  This must be clarified.
  • Which distribution is used in the variation of one standard deviation? linear?(line 245) If you use random values you need to decide on the distribution and this needs to be mentioned.
  • Frequency of inductance measurement? Please note down the frequency or frequency range for the inductance characterization.

 (The frequency does define the extent of eddy current and proximity effect in the copper volume. The sensitivity to dimensional uncertainty of the copper cross-section is expected to depend on the current distribution in the copper. Therefore, it might change over frequency.  In this case, the core material of the copper trace does not contribute to conducting the current and so changes in cross-section by trace have a reduced effect. The proximity effect does also increase the resistance/losses of the coil by multiples of the DC value for higher frequencies.)

Specific comments:

In figure 11, the distribution looks quite non-symmetric, which could be an indicator of the non-linear influence of this factor. Since the paper assumes linear sensitivity of the influencing factor, this could mean the range of variation is pretty high for the linear uncertainty model.

For additional uncertainty sources: The used models for the coil inductance are usually based on homogeneous current distributions in the wire or summarising them in a single current filament. However, due to the proximity effect, the current does concentrate at certain edges of the copper cross-section for higher frequencies. It basically means the current does not align with the copper distribution. The position of the edges is getting more important in this case. PCB-based coils are the worst in this aspect compared to wire-based coils. This part could be considered more, especially in regards to the manufacturing process and the trace undercutting. However, I don't know, if the frequencies for the characterization are high enough to trigger the proximity effect. Typically indicator of the strong proximity effect is the increase of the resistance to a multiple of the DC resistance.

Author Response

Thank you very much for the constructive comments. Following are the itemized responses to the comments.

1. Is the random variation individual or homogeneous?

  All parameter were varied individually. Some of the parameters are best represented by homogeneous variation (e.g. layer distance). However, it is important to maintain the same variability for all factors. This point is added to the revised manuscript (line 249-254 of the revised manuscript).

2. The characteristics of distributions related to the parameters.

It is assumed that all distributions are normal, even though at least one distribution is far from normal (layer-layer misalignment). However, the effect of of layer-layer misalignment is not dominant. We assumed that the error introduced by the nonlinearity of the distribution is not significant.

3. Proximity effect.

Since SRF is around 1 MHz, the skin depth is in the range of 66 µm. Since the cross-sectional area of a single trace is 35 µm × 102 µm, we assumed that the proximity effect is negligible. This is corroborated by the fact the inductance calculated by the model is slightly larger than the average of measurements. If the proximity effect is significant, the measured inductance will be higher than the calculations (line 230-234 of the revised manuscript).

4. Nonlinearity of distribution

As the reviewer have correctly pointed out, the distribution related to layer-layer misalignment is not symmetric. Therefore, the uncertainty analysis based on linearity may have some error. However, layer-layer misalignment is not dominant factor. We assumed that the error due to the nonlinearity of distribution is not significant. This point is added in the revision (line 278-280).

5. Revise conclusions

Indeed, the conclusion section in the original manuscript is rather short. We have highlighted the main contributions and the outlook in the revision.

Thank you very much again for very helpful comments.

Reviewer 2 Report

General Comments

Overall, the paper is strong. The English is adequate, although it could be improved.

Specific Comments

Page 2: In Eq. (2), why is the frequency equal to sqrt(1/LC) and not sqrt(1/LC – R^2/4L^2)? If you solve the equation L d^2I/dt^2 + R dI/dt + I/C = 0, you get the more complicated expression for the frequency, and it is only equal to sqrt(1/LC) if R is very small. Is R small enough that this term can be neglected? I expect this issue would not affect the results much, but I am not sure.

Page 4: The text states that the coils could be calibrated, but that “additional calibration increases the overall complexity and cost of the sensor and mitigates the advantage of using PCB techniques.” In the discussion, the authors might consider describing qualitatively what would be involved in calibration. It is not clear to me that calibrating each coil would be more difficult and time consuming than the effort that would have to be expended making these coils uniform. Some insight into the relative advantages and disadvantages of calibration would be appreciated.

Page 11–12: In Figs. 8–11, I cannot read the axis labels.

Author Response

Thank you very much for the constructive comments. Following are the responses to the comments.

1. Why was the resistance not considered in the derivation of SRF?

You are absolutely correct that the resistance must be considered. However, it can be neglected if the Q factor is high, which is the usual case. The text is modified in the revision (line 68-71)

2. Elaborate on the calibration procedure

In Conclusion, a brief remark has been added regarding the calibration procedure. My collaborators have used a precision target having several different gaps. This will partly automate the calibration procedure. Still, it is quite time-consuming. Since the calibration procedure is very specific, we feel that it is not necessary to delve into the details of the process.

3. Axis labels are not clear

I am not sure why the manuscript file does not show the labels more clearly. When I create the PDF file, the figures are very clear, as they are resolution independent. For your reference, I am attaching the revised manuscript with these responses.

Reviewer 3 Report

The paper deals with the uncertainties due to manufacturing deviations in printed-circuit board sensors.

The authors use image processing techniques to identify variations in the coils’ shape and investigate how each kind of problem influences its inductance.

The text is well written, organized and the contribution is clear. However, some grammar mistakes could be corrected.

The methods to obtain the inductance have been validated by comparing their results with some references. However, I believe that numerical simulations with finite element methods would enhance the paper’s credibility.

In the following, I list some suggestions that I hope would improve the article's final version.

The model expressed by Eq. (1) considers the parameters are lumped. Have you tried to use a model with distributed parameters?

Figure 2 is separated into several figures. I recommend naming a new figure each time you use a different caption. Otherwise, you could place all figures together within the same caption.

In Figure 5, I can see that the noise (salt and pepper) is eliminated. But some objects have been eliminated as well, as described in lines 145-148. I did not understand how the objects cut by the edges are inserted again in the analysis. Explaining it in a few words could be helpful.

Lines 208-210 and 219-222. I suggest replacing both lists with a figure with two flowcharts. It would help to identify the procedures and to point out your contribution.

Line 285. Something is missing there.

I believe more attention could be dealt with the conclusions. It seems you are resuming the paper instead of concluding. I recommend exploring your results to claim the importance of your findings and highlight the importance of your contribution.

Author Response

Thank you very much for very constructive comments. Following are the responses to the comments.

1. Use distributed-parameter model for sensor impedance.

You are correct that the distributed-parameter model is more accurate than the lumped parameter model. However, we are interested in the variability of the inductance. Eq. (1) is shown to illustrate the principle of eddy current sensor, but is not used to assess the variability of the inductance.

2. Separate Fig. 2 into separate figures.

Done.

3. How are the traces at the boundary of an image considered in the analysis?

Since the image analysis focus on the size of cross-sectional areas and the distances between traces, it is important to identify the centroid of each trace. If one trace is at the edge of an image, the centroid will not the center of the trace. Therefore, we eliminated those traces. There are quite a lot of traces that are not at the boundaries for analysis.

4. Replace the lists with flowcharts.

Thank you very much for the suggestion. We have added two figures replacing the lists in the revision, following your suggestion.

5. Use of FEM for numerical validation

Finite-element analysis would be of course more accurate to compute the inductance of the sensing coils. However, it would take very long to do parametric study (10,000 cases in our study). Since the inductance model is validated quite extensively in the references (e.g. ref [7] and [8]), we feel that the model is as accurate as our objectives are concerned.